# Efficient and Thrifty Voting by Any Means Necessary

**Debmalya Mandal**
Columbia University
dm3557@columbia.edu

**Ariel D. Procaccia**
Carnegie Mellon University
arielpro@cs.cmu.edu

**Nisarg Shah**
University of Toronto
nisarg@cs.toronto.edu

**David P. Woodruff**
Carnegie Mellon University
dwoodruf@cs.cmu.edu

## Abstract

We take an unorthodox view of voting by expanding the design space to include both the *elicitation rule*, whereby voters map their (cardinal) preferences to votes, and the *aggregation rule*, which transforms the reported votes into collective decisions. Intuitively, there is a tradeoff between the communication requirements of the elicitation rule (i.e., the number of bits of information that voters need to provide about their preferences) and the efficiency of the outcome of the aggregation rule, which we measure through *distortion* (i.e., how well the utilitarian social welfare of the outcome approximates the maximum social welfare in the worst case). Our results chart the Pareto frontier of the communication-distortion tradeoff.

## 1 Introduction

AI systems are increasingly being used to make decisions that have an impact on people and society. There is much discussion of ways to ensure that such systems reflect appropriate societal values, but it is often unclear what the right choices are [1]. A promising direction is to design systems that incorporate and aggregate people's opinions, by building on work in social choice theory [2, 3, 4].

While the origins of the field can be traced back to the contributions of Condorcet [5] and others in the 18th Century, it was founded in its modern form in the 20th Century. With his famous impossibility result, Arrow [6] pioneered the axiomatic approach to voting, in which voting rules that aggregate ranked preferences of individuals are compared qualitatively based on the axiomatic desiderata they satisfy or violate. This approach underlies most of the work on voting in social choice theory [see, e.g., 7, 8].

By contrast, research in computational social choice [9] has put more emphasis on *quantitative* evaluation of voting rules. In particular, Procaccia and Rosenschein [10] introduced the *implicit utilitarian voting* framework, in which it is assumed that individuals (a.k.a. voters) have underlying cardinal utilities for the different alternatives, and express ranked preferences that are consistent with their utilities. The goal is to choose an alternative that maximizes (utilitarian) *social welfare* — the sum of utilities — by relying on the reported rankings as a proxy for the latent utilities. Specifically, voting rules are compared by their *distortion*, which is the worst-case ratio of the maximum social welfare to the social welfare of the alternative they choose. The implicit utilitarian voting approach has received significant attention in the past decade [11, 12, 13, 14, 15, 16, 17, 18, 19, 20, 21, 22, 23, 24, 25], and voting rules based on it have been deployed on the online voting platform robovote.org.

Benadè et al. [13] observe that implicit utilitarian voting has another advantage: it allows comparing not only voting rules that aggregate ranked preferences, but also voting rules that aggregate other types of ballots, which they refer to as *input formats*. They further argue that we can associate each input format with the best rule for aggregating votes in that format, and ultimately compare the

input formats themselves based on the lowest distortion they make possible. They also introduce a new input format, *threshold approval*, whereby each voter is asked to report whether her utility for each alternative is above or below a given threshold; this input format allows achieving logarithmic distortion. The results of Benadè et al. [13] beg the question: why should we set only a single threshold? What if we set two thresholds and ask each voter to report whether her utility for each alternative is below the lower threshold, between the two thresholds, or above the higher threshold? What if we set five thresholds? Or a million for that matter? Intuitively there is a tradeoff between the number of thresholds and the distortion that can be achieved. However, perhaps adding thresholds is not the most efficient way to drive down distortion; there may be other input formats that encapsulate more useful information. (Spoiler alert: this is indeed the case.)

Our goal in this paper is to characterize the *optimal* tradeoff between elicitation and distortion. Ranking always asks voters to rank their alternatives and always asks for the same amount of information from the voters. On the other hand, consider the input format threshold approval. The larger the number of thresholds, the finer the information we elicit about voter utilities, and the lower the distortion. This is an example of a tradeoff between the amount of information voters are required to report, and the distortion that can be achieved. As we elicit more information from voters about their utilities, we should be able to achieve lower distortion. But exactly how low? To answer this question, we need a precise way to reason about the complexity of vote elicitation. We use the nomenclature of communication complexity [26], and, in particular, examine the number of bits needed to report a vote. Note that this is simply the logarithm of the number of possible votes that a voter can provide in a given input format. Hence, plurality votes that ask a voter to report which of the $m$ alternatives is her top choice contain $\log m$ bits of information, while ranked preferences that ask a voter to rank all $m$ alternatives contain $\log m! = \Theta(m \log m)$ bits of information. Our main research question is this:

*For any $k$, given a budget of at most $k$ bits per vote, what is the minimum distortion any voting rule can achieve?*

## 1.1 Our Results

Before outlining our results, we describe our framework in a bit more detail (a formal model is presented in Section 2). A voting rule $f$ is composed of two parts. Its *elicitation rule* $\Pi_f$ elicits information from voters about their utilities. Essentially, it chooses a (possibly randomized) mapping from utility functions to finitely many (say $k$) possible responses, and each voter uses this mapping to cast her vote. The communication complexity of $f$, denoted $\mathrm{C}(f)$, is then $\mathbb{E}[\log k]$, where the expectation is due to random choices made by $\Pi_f$. The *aggregation rule* $\Gamma_f$ aggregates the votes cast by voters to choose a single alternative (possibly in a randomized way). The distortion of $f$, denoted $\mathrm{dist}(f)$, is the worst-case ratio of the maximum social welfare to the (expected) social welfare of this chosen alternative. The distortion is typically a function of the number of alternatives $m$. Our goal is to study the tradeoff between $\mathrm{C}(f)$ and $\mathrm{dist}(f)$.

Figure 1 shows our results and positions them in the context of prior work. We note that any upper bound with deterministic elicitation (resp. aggregation) also serves as an upper bound with randomized elicitation (resp. aggregation), and the converse holds for lower bounds. For deterministic elicitation, it is known that plurality voting rule achieves $\Theta(m^2)$ distortion with deterministic aggregation and $\log m$ communication complexity, and that it is trivial to achieve $\Theta(m)$ distortion with randomized aggregation and zero communication complexity [20]. Our lower bounds from Section 4 establish that these are the best possible asymptotic bounds with communication complexity at most $\log m$. We show that these bounds do not hold for randomized elicitation by constructing a new voting rule in Section 3, RANDSUBSET, which uses randomized elicitation and achieves $o(m)$ distortion with $o(\log m)$ communication complexity.

We also propose a family of voting rules, PREFTHRESHOLD, which use deterministic elicitation and aggregation, and can achieve $d$ distortion with $O(m \log(d \log m)/d)$ communication complexity, shown as the solid line in Figure 1. This is an improvement over bounds achievable by existing deterministic elicitation methods (even with optimal randomized aggregation): threshold approval voting has distortion $\Omega(\sqrt{m})$ with communication complexity $m$ [13], and ranked voting has distortion $\Omega(\sqrt{m})$ with communication complexity $m \log m$ [12]. In fact, this is also an improvement over the randomized elicitation version of threshold approval voting, which still has distortion $\Omega(\log m/ \log logm)$ with communication complexity $m$ [13].

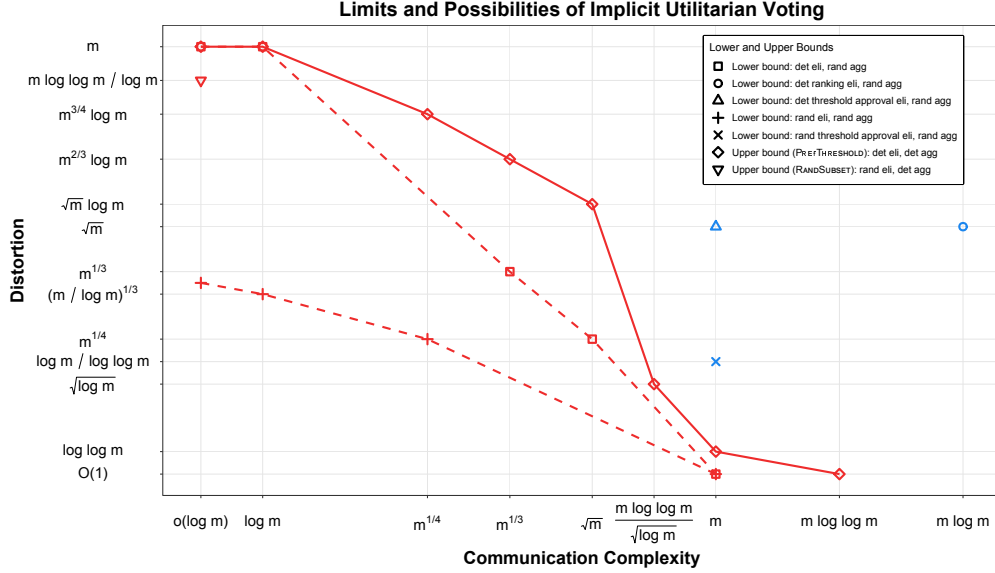

Figure 1: The figure depicts lower and upper bounds on distortion which can be achieved as a function of communication complexity. We use the following abbreviation: det = deterministic, rand = randomized, eli = elicitation, agg = aggregation. Red dashed lines show our lower bounds (Theorems 4 and 6), which apply to all voting rules using deterministic or randomized elicitation. Red diamonds (solid line) and inverted triangle show some of the upper bounds achieved by two families of rules we introduce — PREFTHRESHOLD (Theorem 1) and RANDSUBSET (Theorem 2) — which use deterministic and randomized elicitation, respectively, and deterministic aggregation. We obtain tradeoffs which Pareto-dominate the best possible tradeoffs that existing elicitation methods (shown in blue) — such as threshold approval voting [13] and ranked voting [12] — allow even with randomized aggregation.

In Section 5, we leverage tools from multi-party communication complexity to show that the bounds achieved by PREFTHRESHOLD are nearly optimal: any voting rule with $d$ distortion must have $\Omega(m/d^2)$ communication complexity with deterministic elicitation and $\Omega(m/d^3)$ communication complexity with randomized elicitation. These are presented as dashed lines in Figure 1. Note that our upper and lower bounds differ by a factor that is almost linear or almost quadratic in $d$, and sublogarithmic in $m$. This implies a surprising fact: when our goal is to achieve near-constant distortion, randomization cannot significantly help.

## 1.2 Related Work

There are two threads of research on implicit utilitarian voting. The first thread does not make any assumptions on utilities, other than that they are normalized [10, 11, 12, 13, 19, 20, 24, 25]. The second thread assumes that utilities are induced by a metric [14, 15, 16, 17, 18, 21, 22, 23]; this structure generally enables lower distortion. Our approach is consistent with the former thread.

In addition to the work of Benadè et al. [13], discussed above, an especially relevant paper is that of Caragiannis and Procaccia [11]. Their goal is also to achieve low distortion while keeping the communication requirements low. To that end, they employ specific voting techniques such as approving a single alternative (like plurality) or approving a subset of alternatives (like approval voting) — these require $\log m$ and $m$ bits per voter, respectively — but use what they call an *embedding* to describe how voters translate their cardinal preferences into votes. However, the key difference between the work of Caragiannis and Procaccia and our work is that our design space is much larger: we simultaneously optimize both the embedding and the voting technique (together, these form our elicitation rule), as well as the aggregation rule.[1]

Further afield, Conitzer and Sandholm [27] study how much information about the voters' ranked preferences has to be elicited in order to compute the outcome under a given voting rule. By contrast, we are interested in designing the voting rule, and the very way in which preferences are represented, in order to minimize distortion. In addition, the voting rules we design ask voters to report their approximate utility for their top few choices or for a randomly chosen subset of alternatives. Related ideas have been explored previously [28] or in parallel [29] in the computational social choice literature, albeit in fundamentally different models.

Another loosely related line of work was initiated by Balcan and Harvey [30] and Badanidiyuru et al. [31]. Their goal is to sketch *combinatorial* valuation functions, that is, to encode such functions using a polynomial number of bits in a way that the value of each subset can be recovered approximately. We deal with much simpler valuation functions, but, on the other hand, are looking to achieve much lower communication complexity. We also note that in several query models it is standard to directly query a real number [30, 31, 32, 33, 34]; by contrast, in our framework, asking for even a single real number leads to infinite communication complexity.

## 2 Model

For $k \in \mathbb{N}$, define $[k] = \{1, \ldots, k\}$. Let $x \sim D$ denote that random variable $x$ has distribution $D$. Let $\log$ denote the logarithm to base 2, and $\ln$ denote the logarithm to base $e$. There is a set of alternatives $A$ with $|A| = m$, and a set of voters $N = [n]$. Each voter $i \in N$ is endowed with a valuation $v_i : A \to \mathbb{R}_+$, where $v_i(a) \geq 0$ represents the value of voter $i$ for alternative $a$. Equivalently, we view $v_i \in \mathbb{R}_+^m$ as a vector which contains the voter's value for each alternative. We slightly abuse notation and let $v_i(S) = \sum_{a \in S} v_i(a)$ for $S \subseteq A$. Collectively, voter valuations are denoted $\vec{v} = (v_1, \ldots, v_n)$. Given $\vec{v}$, the (utilitarian) social welfare of an alternative $a$ is $\mathrm{sw}(a, \vec{v}) = \sum_{i \in N} v_i(a)$. Our goal is to elicit information about voter valuations and use it to find an alternative with high social welfare.

**Valuations:** We adopt the standard normalization assumption that $\sum_{a \in A} v_i(a) = 1$ for each $i \in N$. This can be thought of as a "one voter, one vote" principle for cardinal valuations as it prevents voters from overshadowing other voters by expressing very high values. [2] Let $\Delta^m$ denote the $m$-simplex, i.e., the set of all vectors in $\mathbb{R}_+^m$ whose coordinates sum to 1. Hence, we have that $v_i \in \Delta^m$ for each $i \in N$. Given such a vector $v_i \in \Delta^m$, let $\mathrm{supp}(v_i) \subseteq A$ denote the *support* of $v_i$, i.e., the set of alternatives $a$ for which $v_i(a) > 0$.

**Query space:** Consider any interaction with voter $i$ which elicits finitely many bits of information and in which the voter responds deterministically. In this interaction, the voter must provide one of finitely many (say $k$) possible responses. We say that this interaction elicits $\log k$ bits of information.[3] It effectively partitions $\Delta^m$ into $k$ *compartments*, where the compartment corresponding to each response is the set of all valuations which would result in the voter choosing that response. In other words, any interaction which elicits $\log k$ bits of information is equivalent to a *query* which partitions $\Delta^m$ into $k$ compartments and asks the voter to pick the compartment in which her valuation belongs.

Let $\mathcal{Q}$ denote the set of all queries which partition $\Delta^m$ into finitely many compartments. For a query $q \in \mathcal{Q}$, let $k(q)$ denote the number of compartments created by $q$; the number of bits elicited is $\log k(q)$.[4] This query space incorporates traditional elicitation methods studied in the social choice literature. For instance, plurality votes (which ask voters to report their favorite alternative) use $m$ compartments, $k$-approval votes (which ask voters to report the set of their $k$ favorite alternatives) use $\binom{m}{k}$ compartments, threshold approval votes (which ask voters to approve alternatives for which their value is at least a given threshold) use $2^m$ compartments, and ranked votes (which ask voters to rank all alternatives) use $m!$ compartments.

**Voting Rule:** A voting rule (or simply, a *rule*) $f$ consists of two parts: an *elicitation rule* $\Pi_f$ and an *aggregation rule* $\Gamma_f$. The (randomized) elicitation rule $\Pi_f$ is a distribution over $\mathcal{Q}$, according to which a query $q$ is sampled. Each voter $i$ provides a response $\rho_i$ to this query, depending on her valuation

$v_i$. We say that the elicitation rule is *deterministic* if it has singleton support (i.e., it chooses a query deterministically). The (randomized) aggregation rule $\Gamma_f$ takes voter responses $\vec{\rho} = (\rho_1, \dots, \rho_n)$ as input, and returns a distribution over alternatives. We say that the aggregation rule is *deterministic* if it always returns a distribution with singleton support. Slightly abusing notation, we denote by $f(\vec{v})$ the (randomized) alternative returned by $f$ when voter valuations are $\vec{v} = (v_1, \dots, v_n)$. We measure the performance of $f$ via two metrics.

1. The *communication complexity* of $f$ for $m$ alternatives, denoted $C^m(f) = \mathbb{E}_{q \sim \Pi_f} \log k(q)$, is the expected number of bits of information elicited by $f$ from each voter. We drop $m$ from the superscript when its value is clear from the context.

2. The *distortion* of $f$ for $m$ alternatives, denoted $dist^m(f)$, is the worst-case ratio of the optimal social welfare to the expected social welfare achieved by $f$. Again, we drop $m$ from the superscript when its value is clear from the context. Formally,

$$\text{dist}(f) = \sup_{\vec{v} \in (\Delta^m)^n} \frac{\max_{a \in A} \text{sw}(a, \vec{v})}{\mathbb{E}_{\widehat{a} \sim f(\vec{v})} \text{sw}(\widehat{a}, \vec{v})}.$$

While it is desirable for a voting rule to have low communication complexity and low distortion, typically eliciting more information from voters enables achieving low distortion. Our goal is to understand the Pareto frontier of the tradeoff between communication complexity and distortion.

## 3  Upper Bounds

In this section, we derive upper bounds on the best distortion a voting rule can achieve given an upper bound on its communication complexity (equivalently, this gives an upper bound on the communication complexity required to achieve a given level of distortion). We construct two families of voting rules: PREFTHRESHOLD, which use deterministic elicitation and aggregation, and RANDSUBSET, which convert a given voting rule into one which uses randomized elicitation.

### 3.1  Deterministic Elicitation, Deterministic Aggregation

We begin by designing voting rules which use deterministic elicitation and deterministic aggregation — the most practical combination. Caragiannis et al. [20] show that plurality achieves $\Theta(m^2)$ distortion with $\log m$ communication complexity, and even voting rules that elicit ranked preferences, and thus have $\Theta(m \log m)$ communication complexity, cannot achieve asymptotically better distortion.

We propose a novel voting rule PREFTHRESHOLD$_{t,\ell}$, parametrized by $t \in [m]$ and $\ell \in \mathbb{N}$. It is presented as Algorithm 1. Its elicitation rule asks each voter to report the set of her $t$ most preferred alternatives, and for each alternative in this set, report her approximate value for it by picking one of $\ell + 1$ subintervals of $[0, 1]$. Note that for $t = 1$, we use $\ell$ subintervals of $[1/m, 1]$; this is valid because a voter's value for her most favorite alternative must be at least $1/m$. The aggregation rule is also intuitive: it uses the approximate values to compute an estimated social welfare of each alternative, and picks an alternative with the highest estimated social welfare. The next theorem provides bounds on the communication and distortion of PREFTHRESHOLD$_{t,\ell}$.[5]

**Theorem 1.** *For $t \in [m] \setminus \{1\}$ and $\ell \in \mathbb{N}$, we have*

$$C(\text{PREFTHRESHOLD}_{t,\ell}) = \Theta\left(t \log \frac{m(\ell+1)}{t}\right), dist(\text{PREFTHRESHOLD}_{t,\ell}) = O\left(m^{1+2/\ell}/t\right).$$

*For $t = 1$ and $\ell \in \mathbb{N}$, we have*

$$C(\text{PREFTHRESHOLD}_{1,\ell}) = \log(m\ell), \; dist(\text{PREFTHRESHOLD}_{t,\ell}) = O\left(m^{1+1/\ell}\right).$$

PREFTHRESHOLD$_{t,\ell}$ offers a tradeoff between two parameters, $t$ and $\ell$. Increasing either parameter increases the communication complexity but reduces the distortion. It is easy to see that there is no (asymptotic) benefit of choosing $\ell > \log m$. We make several observations.

- $\boldsymbol{t = 1, \ell = 2}$ gives us subquadratic distortion of $O(m\sqrt{m})$ with just one more bit of elicitation than plurality (i.e. $\log m + 1$ bits).

**ALGORITHM 1:** PREFTHRESHOLD$_{t,\ell}$, where $t \in [m]$ and $\ell \in \mathbb{N}$.
---
**Elicitation Rule:**

- If $t > 1$, create $\ell + 1$ buckets: $B_0 = [0, 1/m^2]$ and $B_p = (1/m^{2-2(p-1)/\ell}, 1/m^{2-2p/\ell}]$ for $p \in [\ell]$.
- If $t = 1$, create $\ell$ buckets: $B_1 = [m^{-1}, m^{-1+1/\ell}]$ and $B_p = (m^{-1+(p-1)/\ell}, m^{-1+p/\ell}]$ for $p \in [\ell] \setminus \{1\}$.
- The query asks each voter $i$ to identify set $S_i^t$ of the $t$ alternatives for which she has the highest value (breaking ties arbitrarily), and for each $a \in S_i^t$, identify bucket index $p_{i,a}$ such that $v_i(a) \in B_{p_{i,a}}$.

**Aggregation Rule:**

- For each $p$, let $U_p$ denote the upper endpoint of bucket $B_p$.
- For each voter $i \in N$ and alternative $a \in A$, define $\widehat{v_i}(a) = U_{p_{i,a}}$ if $a \in S_i^t$ and $\widehat{v_i}(a) = 0$ o.w.
- For an alternative $a \in A$, define the *estimated social welfare* as $\widehat{\mathrm{sw}}(a) = \sum_{i \in N} \widehat{v_i}(a)$.
- Return an alternative with the highest estimated social welfare, i.e., $\widehat{a} \in \arg\max_{a \in A} \widehat{\mathrm{sw}}(a)$.
---

- $t = m^{1-\gamma}, \ell = \log m$ gives us sublinear distortion of $O(m^\gamma)$ (for $\gamma \in (0, 1)$) with polynomial communication complexity of $O(m^{1-\gamma} \log m)$.
- $t = m/\sqrt{\log m}, \ell = \log m$ has distortion $O(\sqrt{\log m})$ with communication $o(m)$, and Pareto-dominates threshold approval voting, which has higher communication complexity of $m$ and higher distortion of $\Omega(\log m / \log\log m)$, even with randomized aggregation [13].
- $t = m, \ell = \log m$ leads to constant distortion with communication $O(m \log\log m)$. By contrast, eliciting ranking leads to higher communication complexity of $\Theta(m \log m)$, and also significantly higher distortion of $\Omega(\sqrt{m})$, even with randomized aggregation [12].

## 3.2 Randomized Elicitation, Randomized Aggregation

We now present a generic approach to designing voting rules with randomized elicitation. Given a voting rule $f$ and an integer $s \leq m$, instead of using $f$ to select one alternative from $A$ directly, we sample $S \subseteq A$ with $|S| = s$ at random and use $f$ to select one alternative from $S$. Recall that for $p \in \mathbb{N}$, $\mathrm{C}^p(f)$ and $\mathrm{dist}^p(f)$ denote the communication complexity and distortion of $f$ for $p$ alternatives, respectively.

Clearly, this approach reduces the communication complexity from $\mathrm{C}^m(f)$ to $\mathrm{C}^s(f)$. Its effect on distortion, however, is more subtle. On the one hand, selecting an alternative from $S$ instead of $A$ results in an inevitable loss of welfare because we can only hope to do as well as the best alternative in $S$. On the other hand, the welfare we achieve is related to the welfare of the best alternative in $S$ via the factor $\mathrm{dist}^s(f)$, which can be significantly lower than $\mathrm{dist}^m(f)$. We show that in some cases, this approach reduces distortion in addition to reducing communication complexity. The key challenge in making this approach work is that we cannot apply $f$ directly to select one alternative from $S$, as the total value of the alternatives in $S$ need not be 1. We circumvent this obstacle by eliciting an approximate value of $v_i(S)$ from each voter $i$, making a number of copies of voter $i$ that is approximately proportional to $v_i(S)$ with each copy now having a total value of 1 for alternatives in $S$, and running $f$ on the resulting instance.

---
**ALGORITHM 2:** RANDSUBSET$(f, s)$, where $f$ is a voting rule and $s \in [m]$
---
**Elicitation Rule:**

- Pick $S \subseteq A$ with $|S| = s$ uniformly at random from among all subsets of $A$ of size $s$.
- Partition $[0, 1]$ into $\lceil \log(4m) \rceil$ buckets as follows: $B_0 = \left[0, \frac{1}{4m}\right]$, $B_j = \left(\frac{2^{j-1}}{4m}, \frac{2^j}{4m}\right]$ for $j \in \lceil \log(4m) \rceil$.
- Ask two reports from each voter $i$:
  1. The bucket index $p_i$ such that $v_i(S) = \sum_{a \in S} v_i(a) \in B_{p_i}$;
  2. A response $\rho_i$ to the elicitation rule of $f$ for the set of alternatives $S$ according to the renormalized valuation $\widehat{v_i}$ defined as $\widehat{v_i}(a) = v_i(a)/v_i(S)$ for each $a \in S$.

**Aggregation Rule:**

- Let $L_p$ denote the lower endpoint of bucket $B_p$ for $p \in \lceil \log(4m) \rceil \cup \{0\}$.
- Run the aggregation rule of $f$ on an input which consists of $4m \cdot L_{p_i}$ copies of $\rho_i$ for each $i \in N$.
---

**Theorem 2.** *For every voting rule $f$ and $s \in [m]$, we have $C^m(\text{RANDSUBSET}(f, s)) = C^s(f) + \log\lceil \log(4m) \rceil$ and $dist^m(\text{RANDSUBSET}(f, s)) \leq \frac{4m}{s} \cdot dist^s(f)$.*

Using $f = \text{PREFTHRESHOLD}_{t,\ell}$ and Theorem 1, we obtain that for $s \in [m]$, $t \in [s]$, and $\ell \in \mathbb{N}$, there is a new voting rule $g = \text{RANDSUBSET}(\text{PREFTHRESHOLD}_{t,\ell}, s)$ with

$$C^m(g) = O\left(t \log(s(\ell+1)/t) + \log\log m\right) \text{ and } dist^m(g) = O\left(m \cdot s^{2/\ell}/t\right).$$

Setting $\ell = \log s$, we get $O(m/t)$ distortion. Then, we set $s = t$ to minimize communication complexity to $O(t \log\log t + \log\log m)$. This is slightly better than using $\text{PREFTHRESHOLD}_{t,\log m}$, which achieves $O(m/t)$ distortion with $O(t \log \frac{m \log m}{t})$ communication complexity. In particular, for $t = O(1)$ this reduces communication complexity by a factor of $\log m / \log\log m$.

An interesting choice is $t = \frac{\log m}{\log\log m}$, which leads to distortion $O\left(m \log\log m / \log m\right) = o(m)$ and communication complexity $O\left(t \log\log t + \log\log m\right) = o(\log m)$. Note that this rule has randomized elicitation but deterministic aggregation. By contrast, we later show that with deterministic elicitation, no voting rule can achieve $o(m)$ distortion with communication complexity at most $\log m$, even when randomized aggregation is allowed (Theorem 4).

# 4 Direct Lower Bounds For Deterministic Elicitation

We now turn to deriving lower bounds on the distortion of a voting rule given an upper bound on its communication complexity (equivalently, this gives a lower bound on the communication complexity required to achieve a given level of distortion). In this section, we focus on deterministic elicitation.

Consider a voting rule $f$ which uses deterministic elicitation and has communication complexity at most $\log k$. Hence, the (deterministic) query of $f$ must partition $\Delta^m$ into at most $k$ compartments. Wthout loss of generality, we can assume that $f$ uses exactly $k$ compartments. This is because if $f$ uses $k'$ compartments where $k' < k$, then we can partition some of its compartments into smaller compartments and derive a new voting rule $g$ which uses exactly $k$ compartments, receives at least the information that $f$ receives from the voters, and simulates the aggregation rule of $f$ to achieve the same distortion. Now, establishing a lower bound on the distortion of $f$ requires analyzing the following game between two players, the voting rule $f$ and the adversary.

1. The voting rule $f$ decides the partition of $\Delta^m$ into $k$ compartments.
2. The adversary decides the response of each voter.
3. The voting rule $f$ picks a winning alternative (or a distribution over winning alternatives, if its aggregation rule is randomized).
4. The adversary picks valuations of voters consistent with their responses in the second step.

We use this framework to derive lower bounds on the distortion of voting rules that use deterministic elicitation. We first focus on deterministic aggregation. Perhaps the simplest such voting rule is plurality, which has $\log m$ communication complexity and achieves $\Theta(m^2)$ distortion. This raises an important question: *What distortion can we achieve with deterministic elicitation, deterministic aggregation, and communication complexity less than $\log m$?* The next lemma shows that the answer turns out to be disappointing.

**Theorem 3.** *Every voting rule that has deterministic elicitation, deterministic aggregation, and communication complexity strictly less than $\log m$ has unbounded distortion.*

Now, plurality has communication complexity $\log m$ and achieves $\Theta(m^2)$ distortion. *Can a different voting rule achieve better distortion using only $\log m$ communication complexity?* Perhaps unsurprisingly, we answer this in the negative. But the proof of this intuitive result is quite intricate.

Further, using randomized aggregation we can trivially achieve $O(m)$ distortion with zero communication complexity (by returning the uniform distribution over alternatives). One may wonder: *How much information do we need from the voters to achieve sublinear distortion?* It is easy to show that eliciting plurality votes is not sufficient. Surprisingly, we show that this holds for *every* $\log m$-bit elicitation. That is, even with randomized aggregation, eliciting $\log m$ bits per voter is asymptotically no better than blindly selecting an alternative uniformly at random!

**Theorem 4.** *Let $f$ be a voting rule with deterministic elicitation and $C(f) \leq \log m$. If $f$ uses deterministic (resp. randomized) aggregation, then $dist(f) = \Omega(m^2)$ (resp. $\Omega(m)$).*

For deterministic aggregation, Theorem 4 shows that eliciting $\log m$ bits per voter is not sufficient to achieve $o(m^2)$ distortion. By contrast, we know from Theorem 1 that we can achieve $O(m)$ distortion by eliciting $O(\log m)$ bits per voter. Similarly, for randomized aggregation, Theorem 4 shows that eliciting $\log m$ bits per voter is not sufficient to achieve $o(m)$ distortion. However, we can achieve $o(m)$ distortion if we are willing to elicit $\omega(\log m)$ bits per voter (Theorem 1),[6] or if we are willing to use randomized elicitation (Theorem 2).

## 5 Lower Bounds Through Multi-Party Communication Complexity

In this section, we leverage tools from the literature on multi-party communication complexity to derive lower bounds for both deterministic and randomized elicitation. Specifically, we derive lower bounds on the communication complexity of voting rules that achieve a given level of distortion. We begin by reviewing existing results on multi-party communication complexity, and then derive new results, which help us prove the desired lower bounds in our voting context.

### 5.1 Setup

In multi-party communication complexity, there are $t$ computationally omnipotent players. Each player $i$ holds a private input $X_i \in \mathcal{X}_i$. The *input profile* is the vector $(X_1, \ldots, X_t)$. The goal is to compute the output of a function $f : \mathcal{X}_1 \times \mathcal{X}_2 \times \ldots \times \mathcal{X}_t \rightarrow \{0, 1\}$ on the input profile.

A shared *protocol* $\Pi$ specifies how the players exchange information among themselves and with the center. We use the *blackboard model*, in which messages written by one player are visible to all other players. Let $\Pi(X_1, \ldots, X_t)$ be the random variable denoting the message transcript generated when all players follow the protocol on input profile $(X_1, \ldots, X_t)$; here, the randomness is due to coin tosses by the players or the protocol. The *communication cost* of $\Pi$, denoted $|\Pi|$, is the maximum length of $\Pi(X_1, \ldots, X_t)$ over all input profiles $(X_1, \ldots, X_t)$ and all coin tosses. Given $\delta \geq 0$, we say that $\Pi$ is a $\delta$-*error protocol* for $f$ if there exists a function $\Pi_{\text{out}}$ such that for every input profile $(X_1, \ldots, X_t)$, $\Pr[\Pi_{\text{out}}(\Pi(X_1, \ldots, X_t)) = f(X_1, \ldots, X_t)] \geq 1 - \delta$. The $\delta$-*error communication complexity* of $f$, denoted $R_\delta(f)$, is the communication cost of the best $\delta$-error protocol for $f$.

### 5.2 Multi-Party Fixed-Size Set-Disjointness

The main ingredient of our proof is a standard problem in multi-party communication complexity called the *multi-party set-disjointness* problem, denoted $\text{DISJ}_{m,t}$. Here, each player $i$ holds an arbitrary set $S_i$ from a universe of size $m$. The goal is to distinguish between two types of inputs.

- NO inputs: The sets are pairwise disjoint, i.e., $S_i \cap S_j = \emptyset$ for all $i \neq j$.
- YES inputs: The sets have a unique element in common, but are otherwise pairwise disjoint, i.e., there exists $x$ such that $S_i \cap S_j = \{x\}$ for all $i \neq j$.

It is promised that the input will be one of these two types (in other words, the protocol is free to choose any output on an input that does not satisfy this promise). Following a series of results [35, 36, 37], Gronemeier [38] and Jayram [39] finally established the optimal lower bound of $\Omega(m/t)$.

We introduce a variant of this problem, which we call *multi-party fixed-size set-disjointness* and denote $\text{FDISJ}_{m,s,t}$. It is almost identical to $\text{DISJ}_{m,t}$, except that we know each player $i$ holds a set $S_i$ of a given size $s$. Our goal is to still determine whether the sets are pairwise disjoint ($S_i \cap S_j = \emptyset$ for all $i \neq j$) or pairwise uniquely intersecting (there exists $x$ such that $S_i \cap S_j = \{x\}$ for all $i \neq j$). We use the lower bound on $R_\delta(\text{DISJ}_{m,t})$ to derive the following lower bound on $R_\delta(\text{FDISJ}_{m,s,t})$.

**Theorem 5.** *For a sufficiently small constant $\delta > 0$ and $m \geq (3/2)st$, $R_\delta(\text{FDISJ}_{m,s,t}) = \Omega(s)$.*

Table 1: Comparison between our lower bounds (Theorem 6) and upper bounds (Theorem 1)

| Distortion | Lower Bounds | | Upper Bound |
|---|---|---|---|
| | Deterministic Elicitation | Randomized Elicitation | |
| $O(m^\gamma)$ | $\Omega(m^{1-2\gamma})$ | $\Omega(m^{1-3\gamma})$ | $O(m^{1-\gamma}\log m)$ |
| $O(\log m)$ | $\Omega\left(m/\log^2 m\right)$ | $\Omega\left(m/\log^3 m\right)$ | $O(m\log\log m/\log m)$ |
| $O(1)$ | $\Omega(m)$ | $\Omega(m)$ | $O(m\log\log m)$ |

### 5.3 Lower Bounds on the Communication Complexity of Voting Rules

We now use our lower bound on the $\delta$-error communication complexity of $\text{FDISJ}_{m,s,t}$ to derive a lower bound on the communication complexity of a voting rule in terms of its distortion. We derive different bounds depending on whether the elicitation rule of $f$ is deterministic or randomized. For randomized elicitation, our bound is weaker.

The key insight in the proof is that we can use a voting rule $f$ with $\text{dist}(f) \leq t/2$ to construct a $\delta$-error protocol for solving $\text{FDISJ}_{m,s,t}$, and hence we can use the lower bound on $R_\delta(\text{FDISJ}_{m,s,t})$ from Theorem 5 to derive a lower bound on $\text{C}(f)$. At a high level, consider an instance $(S_1, \ldots, S_t)$ of $\text{FDISJ}_{m,s,t}$. We ask each player $i$ to respond to the query of $f$ according to an artificial valuation function constructed using $S_i$. We then use these responses to create an input for the aggregation rule of $f$. We show that by asking each player an additional question about the alternative returned by the aggregation rule, and possibly running this process a number of times, we can solve $\text{FDISJ}_{m,s,t}$.

**Theorem 6.** *Consider a voting rule $f$ with elicitation rule $\Pi_f$ and $dist(f) = d$. If $\Pi_f$ is deterministic, then $C(f) \geq \Omega\left(m/d^2\right)$, and if $\Pi_f$ is randomized, then $C(f) \geq \Omega\left(m/d^3\right)$.*

Finally, Table 1 summarizes our upper and lower bounds for some special cases. Achieving sublinear distortion makes polynomial communication complexity both necessary (even with randomized aggregation) and sufficient (even with deterministic aggregation). If $\text{dist}(f) = O(\log m)$, our upper and lower bounds differ by only polylogarithmic factors. And for constant distortion, they differ by only a sublogarithmic factor.

## 6 Discussion

We initiated a formal study of the communication-distortion tradeoff in voting, but our work leaves many open questions. The most immediate direction is to close the gap between our upper and lower bounds. Regarding our upper bounds, both families of voting rules that we introduce — PREFTHRESHOLD and RANDSUBSET — use deterministic aggregation, and we do not have better upper bounds using randomized aggregation. Our lower bounds from Theorem 6 are also identical for deterministic and randomized aggregation. This raises an elegant question: *Can randomized aggregation help?* Also, using randomized elicitation in RANDSUBSET, we can achieve sublinear distortion with communication complexity at most $\log m$; Theorem 4 shows that this is not possible with deterministic elicitation. This raises another elegant question: *What is the best possible distortion with randomized elicitation and communication complexity at most $\log m$?* It would also be interesting to improve upon our lower bounds in Section 5, potentially by using a different problem from the multi-party communication complexity literature.

Taking a broader viewpoint, we can consider more general forms of elicitation, like *non-uniform* questions across voters, and questions *adaptive* to past responses. One can also explore the effect of imposing other restrictions on the voting rule such as truthfulness [25, 40]. On a conceptual level, perhaps the main take-away message of our paper is that it pays off to elicit and aggregate preferences "by any means necessary," that is, potentially through highly nonstandard aggregation and, especially, elicitation rules. In the setting of Caragiannis and Procaccia [11], voters are software agents, and this is natural. But when voters are people, it is crucial to understand the implications of such unconventional approaches, both in terms of how communication complexity corresponds to cognitive burden, and in terms of the interpretability and transparency of aggregation rules.

## Acknowledgments

Mandal was partially supported by the Post-Doctoral fellowship from the Columbia Data Science Institute, and part of this work was done while he was a graduate student at Harvard University. Procaccia was partially supported by the National Science Foundation under grants IIS-1350598, IIS-1714140, CCF-1525932, and CCF-1733556; by the Office of Naval Research under grants N00014-16-1-3075 and N00014-17-1-2428; by a J.P. Morgan AI Research Award; and by a Guggenheim Fellowship. Shah was partially supported by the Natural Sciences and Engineering Research Council under a Discovery grant. Woodruff was partially supported by the National Science Foundation under grant CCF-1815840, and part of this work was done while he was visiting the Simons Institute for the Theory of Computing.

## Footnotes

[1]That said, in this work we focus only on deterministic embeddings. That is, we study elicitation rules in which voters deterministically translate their cardinal preferences into votes, and show that the foregoing result is impossible to achieve in this case. We discuss implications of randomized embeddings in Section 6.

[2] Effectively, voters can only report the intensity of their relative preference for one alternative over another.

[3] For a multi-round interaction, we can concatenate the voter's responses in different rounds. This is equivalent to a single-round interaction in which the voter is asked to provide the entire string upfront.

[4] Note that the number of bits elicited may not be an integer, but 2 raised to the power of the number of bits must be an integer. We could take the ceiling to enforce an integral number of bits, and this would only minimally increase elicitation, but some of our lower bounds are sensitive to this non-integral formulation.

[5]All omitted proofs are included in the supplementary material.

[6]For $t = \omega(1)$, PREFTHRESHOLD$_{t,\log m}$ has distortion $O(m/t) = o(m)$ and communication $O(t \log m)$.

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
