[Supplementary Material]

# Supplementary Material

**Debmalya Mandal**
Columbia University
dm3557@columbia.edu

**Ariel D. Procaccia**
Carnegie Mellon University
arielpro@cs.cmu.edu

**Nisarg Shah**
University of Toronto
nisarg@cs.toronto.edu

**David P. Woodruff**
Carnegie Mellon University
dwoodruf@cs.cmu.edu

## 1 Upper Bounds

### 1.1 Deterministic Elicitation, Deterministic Aggregation

**Theorem 1.** *For $t \in [m] \setminus \{1\}$ and $\ell \in \mathbb{N}$, we have*

$$C(\text{PREFTHRESHOLD}_{t,\ell}) = \log\left[\binom{m}{t} \cdot (\ell+1)^t\right] = \Theta\left(t \log \frac{m(\ell+1)}{t}\right),$$

$$dist(\text{PREFTHRESHOLD}_{t,\ell}) = O\left(m^{1+2/\ell}/t\right).$$

*For $t = 1$ and $\ell \in \mathbb{N}$, we have*

$$C(\text{PREFTHRESHOLD}_{1,\ell}) = \log(m\ell), \ dist(\text{PREFTHRESHOLD}_{t,\ell}) = O\left(m^{1+1/\ell}\right).$$

*Proof.* It is evident that the number of possible responses that a voter can provide under $\text{PREFTHRESHOLD}_{t,\ell}$ is $\binom{m}{t} \cdot (\ell+1)^t$ if $t > 1$, and $m\ell$ if $t = 1$. Taking the logarithm of this gives us the desired communication complexity.

We now establish the distortion of $\text{PREFTHRESHOLD}_{t,\ell}$. Let $\vec{v} = (v_1, \ldots, v_n)$ be the underlying valuations of voters. For alternative $a \in A$, recall that $\text{sw}(a, \vec{v}) = \sum_{i \in N} v_i(a)$, and

$$\widehat{\text{sw}}(a) = \sum_{i \in N} \widehat{v_i}(a) = \sum_{i \in N : a \in S_i^t} \widehat{v_i}(a) = \sum_{i \in N : a \in S_i^t} U_{p_{i,a}}.$$

Let $\widehat{a} \in \arg\max_{a \in A} \widehat{\text{sw}}(a)$ be the alternative chosen by the rule, and let $a^* \in \arg\max_{a \in A} \text{sw}(a, \vec{v})$ be an alternative maximizing social welfare.

We begin by finding an upper bound on $\text{sw}(a^*, \vec{v})$ in terms of $\widehat{\text{sw}}(\widehat{a})$.

$$\text{sw}(a^*, \vec{v}) = \sum_{i \in N} v_i(a^*) = \sum_{i \in N : a^* \in S_i^t} v_i(a^*) + \sum_{i \in N : a^* \notin S_i^t} v_i(a^*)$$

$$\leq \sum_{i \in N : a^* \in S_i^t} v_i(a^*) + \sum_{i \in N : a^* \notin S_i^t} \left(\frac{\sum_{a \in S_i^t} v_i(a)}{t}\right)$$

$$\leq \sum_{i \in N : a^* \in S_i^t} \widehat{v_i}(a^*) + \frac{\sum_{a \in A \setminus \{a^*\}} \sum_{i \in N : a^* \notin S_i^t \wedge a \in S_i^t} \widehat{v_i}(a)}{t}$$

$$\leq \widehat{\text{sw}}(a^*) + \frac{\sum_{a \in A \setminus \{a^*\}} \widehat{\text{sw}}(a)}{t} \leq \widehat{\text{sw}}(\widehat{a}) + \frac{(m-1) \cdot \widehat{\text{sw}}(\widehat{a})}{t} = \frac{m+t-1}{t} \cdot \widehat{\text{sw}}(\widehat{a}),$$

$$\tag{1}$$

where the third transition holds because for every $i \in N$ with $a^* \notin S_i^t$ and every $a \in S_i^t$, we have $v_i(a^*) \leq v_i(a)$; the fourth transition holds because for every $i \in N$ and $a \in S_i^t$, $v_i(a) \leq \widehat{v}_i(a)$; the fifth transition follows from the definition of $\widehat{\text{sw}}$; and the sixth transition holds because $\widehat{a}$ is a maximizer of $\widehat{\text{sw}}$.

We now establish the distortion for $t > 1$. The first step is to derive an upper bound on $\widehat{\text{sw}}(\widehat{a})$ in terms of $\text{sw}(\widehat{a}, \vec{v})$. Our bucketing implies that for all $i \in N$ and $a \in S_i^t$, we have $v_i(a) \leq \widehat{v}_i(a) \leq m^{2/\ell}v_i(a) + \frac{1}{m^2}$. Using this, we can derive the following.

$$\widehat{\text{sw}}(\widehat{a}) = \sum_{i \in N : \widehat{a} \in S_i^t} \widehat{v}_i(\widehat{a}) \leq \sum_{i \in N : \widehat{a} \in S_i^t} \left( m^{2/\ell}v_i(\widehat{a}) + \frac{1}{m^2} \right) \leq m^{2/\ell}\text{sw}(\widehat{a}, \vec{v}) + \frac{n}{m^2}. \qquad (2)$$

Next, we derive a lower bound on $\widehat{\text{sw}}(\widehat{a})$, which helps establish a lower bound on $\text{sw}(\widehat{a}, \vec{v})$. Note that for each voter $i \in N$, $\sum_{a \in S_i^t} v_i(a) \geq t/m$. Hence,

$$\sum_{a \in A} \widehat{\text{sw}}(a) = \sum_{i \in N} \sum_{a \in S_i^t} \widehat{v}_i(a) \geq \sum_{i \in N} \sum_{a \in S_i^t} v_i(a) \geq \frac{n \cdot t}{m}.$$

Because $\widehat{a}$ is a maximizer of $\widehat{\text{sw}}$, this yields $\widehat{\text{sw}}(\widehat{a}) \geq n \cdot t/m^2$. Substituting this into Equation (2), we get

$$\frac{n}{m^2} + \text{sw}(\widehat{a}, \vec{v}) \cdot m^{2/\ell} \geq \widehat{\text{sw}}(\widehat{a}) \geq \frac{n \cdot t}{m^2} \Rightarrow \text{sw}(\widehat{a}, \vec{v}) \geq \frac{n \cdot (t-1)}{m^2} \cdot m^{-2/\ell} \geq \frac{n}{m^2} \cdot m^{-2/\ell}. \qquad (3)$$

Applying Equations (1), (2), and (3) in this order, we have

$$\frac{\text{sw}(a^*, \vec{v})}{\text{sw}(\widehat{a}, \vec{v})} \leq \frac{m+t-1}{t} \cdot \frac{\widehat{\text{sw}}(\widehat{a})}{\text{sw}(\widehat{a}, \vec{v})} \leq \frac{m+t-1}{t} \cdot \left( m^{2/\ell} + \frac{n}{m^2 \cdot \text{sw}(\widehat{a}, \vec{v})} \right)$$
$$\leq \frac{m+t-1}{t} \cdot \left( m^{2/\ell} + m^{2/\ell} \right) \in O(m^{1+2/\ell}/t).$$

For $t = 1$, we have that for every $i \in N$ and $a \in S_i^t$, $v_i(a) \leq \widehat{v}_i(a) \leq m^{1/\ell}v_i(a)$. Hence, in Equation (2), the additive factor of $n/m^2$ disappears and the multiplicative factor of $m^{2/\ell}$ becomes $m^{1/\ell}$, yielding $\widehat{\text{sw}}(\widehat{a}) \leq \text{sw}(\widehat{a}, \vec{v}) \cdot m^{1/\ell}$. Similarly, Equation (3) becomes $\text{sw}(\widehat{a}, \vec{v}) \geq \frac{n}{m^2} \cdot m^{-1/\ell}$. Following the same line of proof as for the case of $t > 1$, we obtain

$$\frac{\text{sw}(a^*, \vec{v})}{\text{sw}(\widehat{a}, \vec{v})} \leq m \cdot \frac{\widehat{\text{sw}}(\widehat{a})}{\text{sw}(\widehat{a}, \vec{v})} \leq m \cdot m^{1/\ell},$$

which is the desired bound on distortion. $\qquad \square$

## 1.2 Randomized Elicitation, Randomized Aggregation

**Theorem 2.** *For every voting rule $f$ and $s \in [m]$, we have $C^m(\text{RANDSUBSET}(f, s)) = C^s(f) + \log\lceil \log(4m) \rceil$ and $\text{dist}^m(\text{RANDSUBSET}(f, s)) \leq \frac{4m}{s} \cdot \text{dist}^s(f)$.*

*Proof.* Let $\vec{v} = (v_1, \ldots, v_n)$ denote the underlying valuations of voters. First, let us consider a fixed choice of $S \subseteq A$ with $|S| = s$. Due to our bucketing, we have that for every $i \in N$,

$$\frac{v_i(S)}{2} - \frac{1}{4m} \leq L_{p_i} \leq v_i(S). \qquad (4)$$

Recall that in the input to the aggregation rule of $f$, we have $4m \cdot L_{p_i}$ copies of the response $\rho_i$ of voter $i$. Hence, the social welfare function approximated by the aggregation rule of $f$ is given by

$$\forall a \in S, \ \widehat{\text{sw}}(a, \vec{v}) = \sum_{i \in N} 4m \cdot L_{p_i} \cdot \frac{v_i(a)}{v_i(S)} = 4m \sum_{i \in N} v_i(a) \cdot \frac{L_{p_i}}{v_i(S)}.$$

Combining this with Equation (4), we have that for each $a \in S$,

$$\widehat{\mathrm{sw}}(a, \vec{v}) \geq 4m \sum_{i \in N} v_i(a) \cdot \left( \frac{1}{2} - \frac{1}{4m \cdot v_i(S)} \right) = 2m \cdot \mathrm{sw}(a, \vec{v}) - \sum_{i \in N} \frac{v_i(a)}{v_i(S)} \geq 2m \cdot \mathrm{sw}(a, \vec{v}) - n,$$
(5)

as well as

$$\widehat{\mathrm{sw}}(a, \vec{v}) \leq 4m \sum_{i \in N} v_i(a) \cdot 1 = 4m \cdot \mathrm{sw}(a, \vec{v}).$$
(6)

Let $\widehat{a}$ denote the alternative chosen by our rule. Because the distortion of $f$ for choosing an alternative from $S$ is $\mathrm{dist}^s(f)$, we have that $\mathbb{E}[\widehat{\mathrm{sw}}(\widehat{a}, \vec{v})] \geq \max_{a \in S} \widehat{\mathrm{sw}}(a, \vec{v})/\mathrm{dist}^s(f)$. Note that so far, we have fixed $S$. The expectation on the left hand side is due to the fact that even for fixed $S$, $\widehat{a}$ can be randomized if $f$ is randomized.

Next, we take expectation over the choice of $S$, and use the fact that the optimal alternative $a^* \in \arg\max_{a \in A} \mathrm{sw}(a, \vec{v})$ belongs to $S$ with probability $s/m$. We obtain

$$\mathbb{E}[\widehat{\mathrm{sw}}(\widehat{a}, \vec{v})] \geq \frac{\mathbb{E}[\max_{a \in S} \widehat{\mathrm{sw}}(a, \vec{v})]}{\mathrm{dist}^s(f)} \geq \frac{\frac{s}{m} \cdot \widehat{\mathrm{sw}}(a^*, \vec{v})}{\mathrm{dist}^s(f)} \geq \frac{\frac{s}{m} (2m \cdot \mathrm{sw}(a^*, \vec{v}) - n)}{\mathrm{dist}^s(f)},$$
(7)

where the final transition follows from Equation (5). On the other hand, from Equation (6), we have

$$\mathbb{E}[\widehat{\mathrm{sw}}(\widehat{a}, \vec{v})] \leq 4m \, \mathbb{E}[\mathrm{sw}(\widehat{a}, \vec{v})].$$
(8)

Combining Equations (7) and (8), we have that

$$\mathrm{dist}^m(\textsc{RandSubset}(f, s)) = \frac{\mathrm{sw}(a^*, \vec{v})}{\mathbb{E}[\mathrm{sw}(\widehat{a}, \vec{v})]} \leq \frac{\mathrm{sw}(a^*, \vec{v})}{\frac{\mathrm{sw}(a^*, \vec{v})}{2} - \frac{n}{4m}} \cdot \frac{m}{s} \cdot \mathrm{dist}^s(f) \leq \frac{4m}{s} \cdot \mathrm{dist}^s(f),$$

where the final transition uses the fact that $\mathrm{sw}(a^*, \vec{v}) \geq (1/m) \cdot \sum_{a \in A} \mathrm{sw}(a, \vec{v}) = n/m$. This establishes the desired distortion bound. Since each voter answers the query of $f$ for $s$ alternatives and chooses one of $\lceil \log(4m) \rceil$ buckets, we get $\mathrm{C}^m(\textsc{RandSubset}(f, s)) = \mathrm{C}^s(f) + \log\lceil \log(4m) \rceil$, as desired. □

## 2  Lower Bounds

### 2.1  Direct Lower Bounds for Deterministic Elicitation

We start by establishing a straightforward lemma. Recall that for a valuation $v \in \Delta^m$, $\mathrm{supp}(v)$ denotes the support of $v$.

**Lemma 1.** *Let $f$ be a voting rule which uses deterministic elicitation and deterministic aggregation. Let $q^*$ be the query used by $f$. If some compartment of $q^*$ contains two valuations $v^1$ and $v^2$ such that $\mathrm{supp}(v^1) \cap \mathrm{supp}(v^2) = \emptyset$, then the distortion of $f$ is unbounded.*

*Proof.* Suppose compartment $P$ contains valuations $v^1$ and $v^2$ such that $\mathrm{supp}(v^1) \cap \mathrm{supp}(v^2) = \emptyset$. Let $\widehat{a}$ be the alternative returned by $f$ when all voters pick compartment $P$. Pick $t \in \{1, 2\}$ such that $\widehat{a} \notin \mathrm{supp}(v^t)$. Note that $v^t(\widehat{a}) = 0$, but there exists $a^* \in \mathrm{supp}(v^t)$ such that $v^t(a^*) > 0$.

Define voter valuations $\vec{v} = (v_1, \ldots, v_n)$ such that $v_i = v^t$ for each $i \in N$. This yields $\mathrm{sw}(\widehat{a}, \vec{v}) = 0$ and $\mathrm{sw}(a^*, \vec{v}) > 0$, which implies that $f$ must have infinite distortion. □

**Theorem 3.** *Every voting rule that has deterministic elicitation, deterministic aggregation, and communication complexity strictly less than $\log m$ has unbounded distortion.*

*Proof.* We need the following definition. For $a \in A$, we say that the *unit valuation* corresponding to $a$ is the valuation $v^a \in \Delta^m$ for which $v^a(a) = 1$. Let $f$ be a voting rule that has deterministic elicitation and deterministic aggregation, and let $\mathrm{C}(f) < \log m$. Hence, the query used by $f$ must partition $\Delta^m$ into less than $m$ compartments.

Because there are $m$ unit valuations, by the pigeonhole principle there must exist distinct $a, b \in A$ such that $v^a$ and $v^b$ belong to the same compartment. Because $\mathrm{supp}(v^a) \cap \mathrm{supp}(v^b) = \emptyset$, Lemma 1 implies that the distortion of $f$ must be infinite. □

**Theorem 4.** *Let $f$ be a voting rule which uses deterministic elicitation and has $C(f) \le \log m$. If $f$ uses deterministic aggregation, then $dist(f) = \Omega(m^2)$. If $f$ uses randomized aggregation, then $dist(f) = \Omega(m)$.*

*Proof.* Let $f$ be a voting rule which has deterministic elicitation and $C(f) \le \log m$. As argued above, we can assume $C(f) = \log m$ without loss of generality. Hence, the query $q^*$ used by $f$ partitions $\Delta^m$ into $m$ compartments. Let $\mathcal{P} = (P_1, \ldots, P_m)$ denote the set of compartments. If $f$ has unbounded distortion, we are done. Suppose $f$ has bounded distortion.

Due to Lemma 1, each of $m$ unit vectors must belong to a different compartment. Since there are $m$ compartments, we identify each compartment by the unit valuation it contains. For $a \in A$, let $P^a$ denote the compartment containing unit valuation $v^a$. Before we construct adversarial valuations, we need to define *low valuations* and *high valuations*.

*Low valuations*: We say that a valuation $v \in \Delta^m$ is a *low valuation* if $|\text{supp}(v)| = m/5$ and $v(a) = 5/m$ for every $a \in \text{supp}(v)$. Let $\Delta^{m,\text{low}}$ denote the set of all low valuations. Due to Lemma 1, we have

$$v \in \Delta^{m,\text{low}} \cap P^a \Rightarrow a \in \text{supp}(v) \wedge v(a) = \frac{5}{m}. \tag{9}$$

Let $\mathcal{L} = \{P \in \mathcal{P} : P \cap \Delta^{m,\text{low}} \ne \emptyset\}$ be the set of compartments containing at least one low valuation, and $A^{\mathcal{L}} = \{a \in A : P^a \in \mathcal{L}\}$ be the set of alternatives corresponding to these compartments.

We claim that $|A^{\mathcal{L}}| = |\mathcal{L}| \ge 4m/5 + 1$. Suppose for contradiction that $|A^{\mathcal{L}}| \le 4m/5$. Then, $|A \setminus A^{\mathcal{L}}| \ge m/5$. Hence, there exists a low valuation $v \in \Delta^{m,\text{low}}$ such that $\text{supp}(v) \subseteq A \setminus A^{\mathcal{L}}$. Let $a \in A$ be the alternative for which $v \in P^a$. Because $P^a$ contains a low valuation, $a \in A^{\mathcal{L}}$ by definition. Thus, the construction of $v$ ensures $v(a) = 0$. We have $v \in \Delta^{m,\text{low}} \cap P^a$ with $v(a) = 0$, which contradicts Equation (9). Hence, $|A^{\mathcal{L}}| \ge 4m/5 + 1$.

*High valuations*: We say that a valuation $v \in \Delta^m$ is a *high valuation* if $|\text{supp}(v)| = 2$ and $v(a) = 1/2$ for each $a \in \text{supp}(v)$. Let $\Delta^{m,\text{high}}$ denote the set of high valuations. Note that $|\Delta^{m,\text{high}}| = \binom{m}{2}$. Similarly to the case of low valuations, we can apply Lemma 1, and obtain that

$$v \in \Delta^{m,\text{high}} \cap P^a \Rightarrow a \in \text{supp}(v) \wedge v(a) = \frac{1}{2}. \tag{10}$$

For $a \in A$, let $\mathcal{H}^a = \{P \in \mathcal{L} : \exists v \in \Delta^{m,\text{high}} \cap P \text{ s.t. } a \in \text{supp}(v)\}$. In words, $\mathcal{H}^a$ is the set of compartments from $\mathcal{L}$ which contain at least one high valuation $v$ for which $v(a) = 1/2$. Let $A^{\text{high}} = \{a \in A : |\mathcal{H}^a| \ge m/5\}$. We claim that $|A^{\text{high}}| \ge m/6$.

Suppose this is not true. Let $B = |A \setminus A^{\text{high}}|$. Then, $|B| \ge 5m/6$. Consider $a \in B$. Each of the $m-1$ high valuations which contain $a$ in their support must belong to some compartments in $\mathcal{H}^a \cup (\mathcal{P} \setminus \mathcal{L})$. Since $|\mathcal{H}^a| \le m/5 - 1$ for $a \in B$ and $|\mathcal{P} \setminus \mathcal{L}| \le m/5 - 1$, the $m-1$ high valuations containing $a$ in their support are distributed across at most $2m/5 - 2$ compartments. However, due to Lemma 1, a compartment other than $P^a$ can contain at most one high valuation with $a$ in its support. Hence, $P^a$ must contain at least $m - 1 - (2m/5 - 3) = 3m/5 + 2$ high valuations. Thus, we have established that $|B| \ge 5m/6$ and for each $a \in B$, $P^a$ contains at least $3m/5 + 2$ high valuations. Thus, the number of high valuations is at least $(5m/6) \cdot (3m/5 + 2) > m^2/2 > \binom{m}{2}$, which is a contradiction. Thus, we have $|A^{\text{high}}| \ge m/6$.

We are now ready to prove the desired result for both deterministic and randomized aggregation.

*Voter responses*: When responding to the query $q^*$, suppose each compartment $P \in \mathcal{L}$ is picked by a set $N_P$ of $n/|\mathcal{L}|$ voters.

*Deterministic aggregation*: Let $\widehat{a}$ denote the alternative picked by $f$. We claim that $\widehat{a} \in A^{\mathcal{L}}$. If $\widehat{a} \notin A^{\mathcal{L}}$, consider voter valuations $\vec{v}$ such that every voter $i$ picking compartment $P^a \in \mathcal{L}$ has valuation $v_i = v^a$. Since $\widehat{a} \notin A^{\mathcal{L}}$, we have $v_i(\widehat{a}) = 0$ for each $i \in N$, i.e., $\text{sw}(\widehat{a}, \vec{v}) = 0$. Since $\text{sw}(a, \vec{v}) > 0$ for some $a \in A$, $f$ has infinite distortion, which is a contradiction. Thus, we must have $\widehat{a} \in A^{\mathcal{L}}$.

Now, let us construct the voter valuations as follows. Pick a low valuation $\widehat{v} \in P^{\widehat{a}} \cap \Delta^{m,\text{low}}$, which exists because we have established $\widehat{a} \in A^{\mathcal{L}}$. Note that $\widehat{v}(\widehat{a}) = 5/m$. For each $i \in N_{P^{\widehat{a}}}$, let $v_i = \widehat{v}$.

Pick $a^* \in A^{\text{high}} \setminus \{\widehat{a}\}$. Let $\bar{P}$ be the compartment containing the high valuation under which both $\widehat{a}$ and $a^*$ have utility $1/2$. For each $P \in \mathcal{H}^{a^*} \setminus \{P^{\widehat{a}}, \bar{P}\}$, and for each $i \in N_P$, let $v_i$ be the high valuation in $P$ such that $v_i(a^*) = 1/2$ and $v_i(\widehat{a}) = 0$. For every other $P^a \in \mathcal{L}$ and every $i \in N_{P^a}$, let $v_i = v^a$.

Observe that under these valuations, $\text{sw}(\widehat{a}, \vec{v}) = \Theta(n/m^2)$, whereas, since $|\mathcal{H}^{a^*}| \geq m/5$ and $|\mathcal{L}| \leq |\mathcal{P}| = m$, $\text{sw}(a^*, \vec{v}) = \Theta(n)$. We conclude that $\text{dist}(f) = \Omega(m^2)$.

*Randomized aggregation*: Note that $f$ must select at least one alternative $a^* \in A^{\text{high}}$ with probability at most $1/|A^{\text{high}}| \leq 6/m$. Construct voter valuations such that for every $P \in \mathcal{H}^{a^*}$ and every $i \in N_P$, $v_i$ is the high valuation under which $v_i(a^*) = 1/2$. For every $P^a \in \mathcal{L} \setminus \mathcal{H}^{a^*}$, and for every $i \in N_{P^a}$, let $v_i = v^a$. It holds that $\text{sw}(a^*, \vec{v}) = \Theta(n)$ (as before), whereas $\text{sw}(a, \vec{v}) = O(n/m)$ for every $a \in A \setminus \{a^*\}$. Because $f$ selects $a^*$ with probability at most $6/m$, we have $\mathbb{E}_{\widehat{a} \sim f(\vec{v})}[\text{sw}(\widehat{a}, \vec{v})] = O(n/m)$, implying $\text{dist}(f) = \Omega(m)$, as required. $\qquad\square$

## 2.2 Lower Bound for Plurality Votes

In this section, we show that eliciting plurality votes (whereby each voter picks her most favorite alternative) results in $\Omega(m)$ distortion, even with randomized aggregation. This is implied by Theorem 4, which proves this for any elicitation that has at most $\log m$ communication complexity. However, for the special case of plurality votes, we can provide a much simpler proof.

**Theorem 5.** *Every voting rule which elicits plurality votes incurs $\Omega(m)$ distortion.*

*Proof.* For simplicity, let the number of voters $n$ be divisible by the number of alternatives $m$. Consider an input profile in which the set of voters $N$ is partitioned into equal-size sets $\{N_a\}_{a \in A}$ such that for each $a \in A$, $a$ is the most favorite alternative of every voter in $N_a$.

Take any voting rule $f$. It must return some alternative $a^* \in A$ with probability at most $1/m$. Now, construct adversarial valuations of voters $\vec{v}$ as follows.

- For all $i \in N_{a^*}$, $v_i(a^*) = 1$ and $v_i(a) = 0$ for all $a \in A \setminus \{a^*\}$.

- For all $\widehat{a} \in A \setminus \{a^*\}$ and $i \in N_{\widehat{a}}$, $v_i(\widehat{a}) = v_i(a^*) = 1/2$ and $v_i(a) = 0$ for all $a \in A \setminus \{a^*, \widehat{a}\}$.

Under these valuations, we have $\text{sw}(a^*, \vec{v}) \geq n/2$, while $\text{sw}(a, \vec{v}) = (n/m) \cdot (1/2)$ for every $a \in A \setminus \{a^*\}$. Hence, the distortion of $f$ is

$$\text{dist}(f) \geq \frac{\text{sw}(a^*, \vec{v})}{\frac{1}{m}\text{sw}(a^*, \vec{v}) + \frac{m-1}{m}\frac{n}{2m}} = \Omega(m),$$

where the final transition holds when substituting $\text{sw}(a^*, \vec{v}) \geq n/2$. $\qquad\square$

# 3 Lower Bounds Through Multi-Party Communication Complexity

## 3.1 Lower Bound on the Communication Complexity of $\text{FDISJ}_{m,s,t}$

In this section, we prove a lower bound on the communication complexity of multi-party fixed-size set-disjointness. Let us recall Theorem 6.

**Theorem 6.** *For a sufficiently small constant $\delta > 0$ and $m \geq (3/2)st$, $R_\delta(\text{FDISJ}_{m,s,t}) = \Omega(s)$.*

*Proof.* Suppose there is a $\delta$-error protocol $\Pi$ for $\text{FDISJ}_{m,s,t}$. We use it to construct a $2\delta$-error protocol $\Pi'$ for $\text{DISJ}_{m',t'}$, where $m' = st/2$ and $t' = 2t$.

Consider an instance $(S'_1, \ldots, S'_{t'})$ of $\text{DISJ}_{m',t'}$. Due to the promise that the sets are either pairwise disjoint or pairwise uniquely intersecting, we have that at most one of the $m'$ elements can appear in multiple sets. Hence, $\sum_{i=1}^{t'} |S'_i| \leq m' - 1 + t'$. Due to the pigeonhole principle, there must exist at least $t'/2 = t$ sets of size at most $2(m' + t' - 1)/t'$. Note that

$$\frac{2(m' + t' - 1)}{t'} = \frac{st/2 + 2t - 1}{t} = \frac{s}{2} + 2 - \frac{1}{t} \leq s.$$

The final transition holds when $s \geq 4$. When $s < 4$, the lower bound of $\Omega(s)$ is trivial.

Consider a set of $t$ players $\{i_1, \ldots, i_t\}$ such that $|S'_{i_k}| \leq s$ for each $k \in [t]$. Suppose that each such player $i_k$ adds $s - |S'_{i_k}|$ unique elements to $S'_{i_k}$ and creates a set $S_{i_k}$ with $|S_{i_k}| = s$. The number of unique elements required is at most $st$. Hence, the total number of elements used in sets $S_{i_1}, \ldots, S_{i_t}$ is at most $m' + st = (3/2)st \leq m$. In other words, these sets can be created using the $m$-element universe of $\mathrm{FDISJ}_{m,s,t}$. Further, it is easy to check that sets $S_{i_1}, \ldots, S_{i_t}$ are pairwise disjoint (resp. pairwise uniquely intersecting) if and only if sets $S'_1, \ldots, S'_{t'}$ are pairwise disjoint (resp. pairwise uniquely intersecting). Thus, $(S_{i_1}, \ldots, S_{i_t})$ is a valid instance of $\mathrm{FDISJ}_{m,s,t}$ and has the same solution as the instance $(S'_1, \ldots, S'_{t'})$ of $\mathrm{DISJ}_{m',t'}$.

Our goal is to construct a $2\delta$-error protocol $\Pi'$ for $\mathrm{DISJ}_{m',t'}$ that solves $(S'_1, \ldots, S'_{t'})$ by effectively running the given $\delta$-error protocol $\Pi$ for $\mathrm{FDISJ}_{m,s,t}$ on $(S'_{i_1}, \ldots, S'_{i_t})$. We could ask each player $i$ to report a single bit indicating whether $|S'_i| \leq s$, determine $t$ players for which this holds, and then run $\Pi$ on them. However, this would add a $t'$-bit overhead. Instead, we would like to bound the overhead in terms of the communication cost of $\Pi$, denoted $|\Pi|$, which could be significantly smaller.

This is achieved as follows. We first order the players according to a uniformly random permutation $\sigma$. Then, we simulate $\Pi$. Every time $\Pi$ wants to interact with a new player, we ask players that we have not interacted with so far, in the order in which they appear in $\sigma$, whether their sets have size at most $s$, until we find one such player. Then, we let $\Pi$ interact with this player. Protocol $\Pi'$ terminates naturally when protocol $\Pi$ terminates (and returns the same answer), but terminates abruptly if, at any point, it has interacted with more than $2|\Pi|/\delta$ players (and returns an arbitrary answer).

Note that $|\Pi|$ is also an upper bound with the number of players that $\Pi$ needs to interact with. Let $X$ be the smallest index such that there are at least $|\Pi|$ players having sets of size at most $s$ among the first $X$ players in $\sigma$. Then, because at least half of the players have sets of size at most $s$, we have $\mathbb{E}[X] \leq 2 \cdot |\Pi|$. Due to Markov's inequality, we have that $\Pr[X > 2|\Pi|/\delta] \leq \delta$. Hence, the probability that $\Pi'$ terminates abruptly is at most $\delta$. When it does not terminate abruptly, it returns the wrong answer with probability at most $\delta$ (as $\Pi$ is a $\delta$-error protocol). Hence, due to the union bound, we conclude that $\Pi'$ is a $2\delta$-error protocol for $\mathrm{DISJ}_{m',t'}$.

Finally, we have that $|\Pi'| \leq 2|\Pi|/\delta + |\Pi| = |\Pi|(1 + 2/\delta)$. When $\delta$ is sufficiently small, Gronemeier [1] showed that $|\Pi'| \geq R_{2\delta}(\mathrm{DISJ}_{m',t'}) = \Omega(m'/t') = \Omega(s)$. Hence, we have that $|\Pi| = \Omega(s)$. Since this holds for every $\delta$-error protocol $\Pi$ for $\mathrm{FDISJ}_{m,s,t}$, we have $R_\delta(\mathrm{FDISJ}_{m,s,t}) = \Omega(s)$. $\qquad\square$

## 3.2 Lower Bounds on the Communication Complexity of Voting Rules

**Theorem 7.** *For a voting rule $f$ with elicitation rule $\Pi_f$ and $\mathrm{dist}(f) = d$, the following hold.*

- *If $\Pi_f$ is deterministic, then $C(f) \geq \Omega\left(m/d^2\right)$.*

- *If $\Pi_f$ is randomized, then $C(f) \geq \Omega\left(m/d^3\right)$.*

*Proof.* Let $t = 2 \cdot \mathrm{dist}(f)$ and $s = 2m/(3t)$. Note that for these parameters, we have $R_\delta(\mathrm{FDISJ}_{m,s,t}) = \Omega(s)$ from Theorem 6.

Consider an input $(S_1, \ldots, S_t)$ to $\mathrm{FDISJ}_{m,s,t}$ with a universe $U$ of size $m$. Let us create an instance of the voting problem with a set of $n$ voters $N$ and a set of $m$ alternatives $A$. Each alternative in $A$ corresponds to a unique element of $U$. Partition the set of voters $N$ into $t$ equal-size buckets $\{N_1, \ldots, N_t\}$. Here, bucket $N_i$ corresponds to player $i$, and consists of $n/t$ voters that each have valuation $v^{S_i}$ given by $v^{S_i}(a) = 1/s$ for each $a \in S_i$ and $v^{S_i}(a) = 0$ for each $a \notin S_i$. Let $\vec{v}$ denote the resulting profile of voter valuations. Note that under these valuations, $\mathrm{sw}(a, \vec{v}) = \frac{n}{ts}\sum_{i=1}^{t} \mathbb{1}[a \in S_i]$, where $\mathbb{1}$ is the indicator variable. Due to the promise that an element either belongs to at most one set or belongs to every set, we have $\mathrm{sw}(a, \vec{v}) \in \{0, n/(ts), n/s\}$. We say that $a$ is a "good" alternative if $\mathrm{sw}(a, \vec{v}) = n/s$ and a "bad" alternative otherwise.

We define two processes that will help covert our voting rule $f$ into a protocol for $\mathrm{FDISJ}_{m,s,t}$.

*Process E:* In this process, we ask each player $i$ to respond to the query posed by voting rule $f$ (possibly selected in a randomized manner) according to valuation $v^{S_i}$. We note that this requires a total of $t \cdot C(f)$ bits of communication from the players.

*Process A:* We take players' responses from process E, create $n/t$ copies of the response of each player, and pass the resulting profile as input to the aggregation rule $\Gamma_f$ to obtain the returned alternative $\widehat{a}$ (possibly selected in a randomized manner). We end the process by determining if $\widehat{a}$ is a good alternative or a bad alternative. This requires eliciting 2 extra bits of information: we can ask any two players $i$ and $j$ whether their sets contain $\widehat{a}$, and due to the promise of $\mathrm{FDISJ}_{m,s,t}$, we know that $\widehat{a}$ is good if and only if it belongs to both $S_i$ and $S_j$.

Knowing whether $\widehat{a}$ is good or bad is useful for solving the given instance of $\mathrm{FDISJ}_{m,s,t}$ due to the following reason.

1. If $(S_1, \ldots, S_t)$ is a "NO input", then we know that every alternative is a bad alternative. Hence, $\mathrm{sw}(a, \vec{v}) \leq (n/t) \cdot (1/s) = n/(ts)$ for each $a \in A$. In particular, this implies $\mathrm{sw}(\widehat{a}, \vec{v}) \leq n/(ts)$ with probability 1.

2. If $(S_1, \ldots, S_t)$ is a "YES input", then there exists a unique good alternative $a^* \in A$ with $\mathrm{sw}(a^*, \vec{v}) = n/s$, and every other alternative $a$ is a bad alternative with $\mathrm{sw}(a, \vec{v}) \leq n/(ts)$. Because $\mathrm{dist}(f) = t/2$, we have that $\mathbb{E}[\mathrm{sw}(\widehat{a}, \vec{v})] \geq \frac{n/s}{t/2} = \frac{2n}{ts}$. This implies that $\Pr[\mathrm{sw}(\widehat{a}, \vec{v}) = n/s] = \Pr[\widehat{a} = a^*] \geq 1/t$ because if $\Pr[\widehat{a} = a^*] < 1/t$, then $\mathbb{E}[\mathrm{sw}(\widehat{a}, \vec{v})] < (1/t) \cdot (n/s) + 1 \cdot n/(ts) = 2n/(ts)$, which is a contradiction.

We are now ready to use $f$ to construct a protocol for $\mathrm{FDISJ}_{m,s,t}$, and use Theorem 6 to derive a lower bound on $\mathrm{C}(f)$. We consider two cases depending on whether the elicitation rule $\Pi_f$ is deterministic or randomized.

1. *Deterministic elicitation*: In this case, we run process E once and then run process A $t \ln(1/\delta)$ times. In a NO input, we always get a bad alternative. In a YES input, each run of process A returns a good alternative with probability at least $1/t$. Hence, the probability that we get a good alternative at least once is at least $1 - (1 - 1/t)^{t \ln(1/\delta)} \geq 1 - \delta$. Hence, this is a $\delta$-error protocol for $\mathrm{FDISJ}_{m,s,t}$ which requires $t \cdot \mathrm{C}(f) + t \ln(1/\delta) \cdot 2$ bits of total communication from the players. Using Theorem 6, we have that $t \cdot (\mathrm{C}(f) + 2 \ln(1/\delta)) = \Omega(s)$. Using $s = 2m/(3t)$ and $t = 2d$, we have $\mathrm{C}(f) = \Omega(m/d^2)$.

2. *Randomized elicitation*: In this case, we run E once followed by running A once. And we repeat this entire process $t \ln(1/\delta)$ times. Note that we need to repeat process E because the elicitation is also randomized. Like in the previous case, we always get a bad alternative in a NO input, and get a good alternative with probability at least $1/t$ in each run in a YES input. Hence, in a YES input, we get a good alternative in at least one run with probability at least $1 - (1 - 1/t)^{t \ln(1/\delta)} \geq 1 - \delta$. This results in a $\delta$-error protocol for $\mathrm{FDISJ}_{m,s,t}$ which requires $t \ln(1/\delta) \cdot (t \cdot \mathrm{C}(f) + 2)$ bits of total communication from the players. Using Theorem 6, we have $t \ln(1/\delta) \cdot (t \cdot \mathrm{C}(f) + 2) = \Omega(s)$. Using $s = 2m/(3t)$ and $t = 2d$, we have $\mathrm{C}(f) = \Omega(m/d^3)$.

These are the desired lower bounds on $\mathrm{C}(f)$. $\qquad\square$

## References

[1] A. Gronemeier. Asymptotically optimal lower bounds on the NIH-multi-party information complexity of the AND-function and disjointness. In *Proceedings of the 26th International Symposium on Theoretical Aspects of Computer Science (STACS)*, pages 505–516, 2009.