[Reviews · NeurIPS 2019]

Reviewer 1



The (seemingly novel) premise of the paper is that we can gain insight into voting theory by studying it through the premise of communication complexity (in addition to the standard desideratum of social welfare maximization). The paper presents lower bounds, families of new voting rules to provide upper bounds, and classification of previously-studied voting rules in terms of the communication complexity/distortion tradeoff. Overall, the paper seemed original with a comprehensive set of results for a carefully set up and clearly presented model for an important problem. My main criticism is that parts of the intro were hard to follow, and Figure 1 summarizing the results was helpful but could be improved. The specific comments below are suggestions for how to improve Section 1 for readability. - The intro seemed to indicate that the elicitation rule would project real utilities to rankings, which presented an overly narrow view of the range of elicitation rules considered in the paper and made it confusing to understand how to provide bounds on distortion if utilities could be arbitrary. This was clarified in Section 2, but it would help for clarification to come earlier. -Is there a reason RandSubset is not in Fig 1? It would help for comparison. -It would help to clearly classify each voting rule in the legend by deterministic/randomized elicitation/aggregation. -Fig 1 could be improved by adding pointers to the relevant section of the paper for each entry, and it would be nice to include a brief description somewhere of the rules from prior works. -The first par of 1.1 could improve by specifying the domain/range of each rule type. -The second par of 1.1 is quite hard to follow even when simultaneously referencing Fig 1.

Reviewer 2



The results build on a line of research in analyzing the ‘distortion’ of voting rules, which is the ratio of highest social welfare alternative to the selected alternative. This work takes the idea beyond analyzing popular voting rules and asks whether there exists any voting rule that can achieve a desired distortion while minimizing communication, or vice versa. Section 4 & 5 were a little confusing -- the results are organized by proof technique vs. what is proved, e.g. both sections have bounds for deterministic elicitation. Authors should make clear section 4’s results are more meaningful for low-communication voting rules. I have not read the proofs in detail from the supplemental pages, although the intuition conveyed in the paper seem sound. The existence of PrefThreshold as a way to trade-off seems quite interesting and desirable as a voting rule, but as the authors acknowledge it may seem bizarre to ask voters to map their utility to buckets. Nevertheless I find this method to be original and novel. Wrt related works, there may be relationship to complexity of active learning but I cannot articulate a direct relationship at this time. There is a paper seem to cover ‘threshold voting’: https://ieeexplore.ieee.org/abstract/document/370218 this may or may not be directly related to PrefThreshold. Overall the paper is well-written and organized. It does not get too cluttered in mathematical details and try to convey underlying intuition. The biggest concern with this submission is lack of relevance to NeurIPS, which are typically papers addressing dynamics of learning. The problem and proof techniques are not really relevant to learning, and the topic is only of interest to a small community in AI, so this paper may not be appropriate for the NeurIPS audience. ---- Update I raise overall score without considering relevance. Paper makes a great contribution to information complexity of voting.

Reviewer 3



In this paper, the authors study the design of voting rules. The authors consider two aspects of voting rules: i) How well they optimize the social welfare (this is an utilitarian point of view). This is measured by a distortion function (ratio of obtained to optimal social welfare), as is classical to do in related work. ii) How complex the elicitation rule is. This is measured in terms of (expected, since randomization is allowed) communication complexity (how many bits are needed for an agent to report a vote, or equivalently, log of the number of responses available to an agent). The aim of this paper is to characterize the optimal trade-off between distortion and communication complexity. In other words, under a limited communication complexity budget, what is the best social welfare that can be achieved, and how does it compare to the unconstrained social welfare. To do so, the authors provide lower and upper bound on this trade-off, that while not quite matching, are fairly close to each other. Upper bounds are the object of Section 3. There, the authors provide upper bounds for both deterministic and randomized voting rules. To obtain these upper bounds, the authors introduce two new classes of voting rules: i) For deterministic voting rules, the authors consider an algorithm called PrefThreshold that works as follows. The elicitation rule is quite natural: each agent to report their top $t$ alternatives, and to report in what sub-interval (among $l$ of them) their value for each such alternative belongs. The larger the number of alternatives and the finer the decomposition in the sub-intervals get, the better the distortion becomes, but this comes at a higher communication complexity cost. The aggregation rule is also simple and natural: the approximate values from the sub-intervals are used to estimate the social welfare, and the winner is the outcome with the highest estimated SW. To me, this class of algorithm sounds very appealing as it is very natural and could realistically be implemented in practice, via two simple questions: “who are your t favorite candidates?” and “How do you rate them on a scale from 1 to l?” In Theorem 1, the authors characterize the communication complexity and the distortion of the algorithm as a function of $t$ and $l$, then discuss what choices of $t$ and $l$ lead to which trade-off. This allows them to obtain the following upper bounds on what trade-offs are possible. ii) For randomized voting rules, the authors consider a rule that selects a subset of actions at random and only elicits information from the voters about their valuations for this subset. Such randomization in voting rules has probably more restricted applications (I cannot generally imagine current societies accepting randomness in the outcome of legislative or presidential elections, especially when the voting rules randomly throws away part of the candidates – and possibly your top candidate with some probability -- at the beginning), but I think studying what can be achieved by randomized voting rules and how much one can gain compared to deterministic ones is an interesting theoretical question. The authors then develop lower bounds in Sections 4 and 5. The lower bounds of Section 4 apply to deterministic elicitation rules and show that i) if the communication complexity is strictly less than log m, the distortion is unbounded, and ii) even when the aggregation rule is randomized, one cannot hope to get sub-linear distortion when the communication complexity is restricted to exactly log m at most. This shows an interesting gap with the implications of Theorem 2 (this is possible with randomized elicitation rules). Section 5 shows general lower bounds on the communication complexity needed as a function of the distortion in Theorem 6; this gives a lower bound on the trade-off between communication complexity and distortion that applies much more widely (for any $d$) than what was known from previous work. These bounds differ by a factor proportional to the inverse of the distortion between deterministic and randomized voting rules; for small distortion, these complexities only differ by a small factor. They then show how these results apply to special cases; in particular, if one is aiming to have at most logarithmic distortion, then the lower bounds for both deterministic and randomized voting rules are within a logarithmic factor of the upper bounds on computational complexity. I.e., in this regime, the bounds provided by the authors are nearly tight. Put all together, the results paint a fairly complete picture of the trade-off communication complexity in this setting. The results are theoretically solid and interesting and constitute a vast improvement compared to previous work. The only complaint that I have about this paper is that there is no discussion of strategic behavior and truthfulness (could voters manipulate their vote here to get an outcome they prefer, and how much would that impact the social welfare?), but this can be left to future work. To me, this paper is a clear accept.

[Author Response · NeurIPS 2019]

We sincerely thank all the reviewers for their detailed comments, which will be very helpful in improving the paper when we revise it.

**Reviewers 1 & 4: Incentives and truthfulness**

In classical social choice, truthfulness is often seen as a nonstarter due to the Gibbard-Satterthwaite impossiblity theorem, and a subsequent result of Gibbard (1977) that applies to randomized rules. However, these results only hold when voters have (and are asked to report) ranked preferences.

Elicitation of cardinal utilities through truthful *non-direct-revelation* mechanisms (which ask voters to submit votes in some format other than directly report their utilities) is not as well studied and could lead to interesting future work.

**Reviewer 3: Relevance to NeurIPS**

At the risk of being subjective, we would like to point out that the scope of NeurIPS has widened in the last few years, as the conference has become a nexus of AI research. Moreover, the NeurIPS audience has long had an interest in computational social choice. For example, the following computational social choice papers were published in NeurIPS; they do not study the dynamics of learning (nor, for that matter, deal with learning at all), but rather focus on good/optimal decisions in AI systems and beyond.

1. Magdon-Ismail and Xia. "A Mathematical Model for Optimal Decisions in a Representative Democracy." NeurIPS-2018.

2. Procaccia and Shah. "Is Approval Voting Optimal Given Approval Votes?" NeurIPS-2015.

3. Jiang et al. "Diverse Randomized Agents Vote to Win." NeurIPS-2014.

4. Azari Soufiani et al. " A Statistical Decision-Theoretic Framework for Social Choice." NeurIPS-2014 (oral presentation).

5. Azari Soufiani et al. "Random Utility Theory for Social Choice." NeurIPS-2012.

**Reviewer 3: Threshold voting**

The reviewer points to a paper about "threshold voting," but this is a very different idea. The author is interested only in the case where each voter submits a plurality vote, and a "threshold voting method with threshold $t$" aims to find some alternative which is voted for by at least $t$ of the voters. This is less interesting from the computational and informational viewpoints; the author is instead interested in efficient hardware implementations of this method.

[Meta-Review · NeurIPS 2019]

This work provides a comprehensive study of trade-offs between communication complexity and distortion for voting rules, and reviewers were unanimously in favor of acceptance. Please take their constructive feedback into account for the final version, especially with regard to the exposition. While there were concerns as to the fit at NeurIPS given the topic of the paper, the paper is clearly is relevant to AI. (That said, to the extent that the authors can draw connections between their work and AI / ML explicitly, perhaps in the introduction, we would encourage them to do so.)